# Offset Unlearning for Large Language Models

**James Y. Huang**                                                    *huangjam@usc.edu*
*University of Southern California*

**Wenxuan Zhou**                                                      *zhouwenx@usc.edu*
*University of Southern California*

**Fei Wang**                                                         *fwang598@usc.edu*
*University of Southern California*

**Fred Morstatter**                                                      *fred@isi.edu*
*University of Southern California*

**Sheng Zhang**                                                  *shezhan@microsoft.com*
*Microsoft Research*

**Hoifung Poon**                                                *hoifung@microsoft.com*
*Microsoft Research*

**Muhao Chen**                                                     *muhchen@ucdavis.edu*
*University of California, Davis*

**Reviewed on OpenReview:** *https: // openreview. net/ forum? id= A4RLpHPXCu*

## Abstract

Despite the strong capabilities of Large Language Models (LLMs) to acquire knowledge from their training corpora, the memorization of sensitive information in the corpora such as copyrighted, biased, and private content has led to ethical and legal concerns. In response to these challenges, unlearning has emerged as a potential remedy for LLMs affected by problematic training data. However, previous unlearning techniques are either not applicable to black-box LLMs due to required access to model internal weights, or violate data protection principles by retaining sensitive data for inference-time correction. We propose $\delta$-UNLEARNING, an offset unlearning framework for black-box LLMs. Instead of tuning the black-box LLM itself, $\delta$-UNLEARNING learns the logit offset needed for unlearning by contrasting the logits from a pair of smaller models. Experiments demonstrate that $\delta$-UNLEARNING can effectively unlearn target data while maintaining similar or even stronger performance on general out-of-forget-scope tasks. $\delta$-UNLEARNING also effectively incorporates different unlearning algorithms, making our approach a versatile solution to adapting various existing unlearning algorithms to black-box LLMs.

## 1 Introduction

Large Language Models (LLMs) are capable of memorizing a large amount of information derived from their training corpus. While LLMs are empowered by the abundance of knowledge they acquire during training, their training data may contain sensitive information that should not be memorized by LLMs. Previous studies have shown LLMs can reproduce copyrighted materials Chang et al. (2023); Eldan & Russinovich (2023); Karamolegkou et al. (2023), generate harmful and biased content Shaikh et al. (2023), and reveal private information Staab et al. (2024), raising both ethical and legal concerns. The introduction of data protection regulations such as the *right to be forgotten* Hoofnagle et al. (2019); Zhang et al. (2023); Min

| Unlearning Method | Black-box | Privacy |
|---|:---:|:---:|
| Gradient Ascent | ✗ | ✓ |
| Data Relabeling | ✗ | ✓ |
| In-context Unlearning | ✓ | ✗ |
| $\delta$-UNLEARNING | ✓ | ✓ |

Table 1: Comparison with existing unlearning methods. Previous techniques either require access to LLM's internal weights, or retain sensitive information for inference.

et al. (2024) also highlights the need for erasing the influence of problematic data when deploying LLMs in real-world applications.

One potential solution to this challenge is *unlearning*, where the goal is to "forget" a set of training data without hurting the model's performance on out-of-forget-scope tasks. An *exact unlearning* approach would require retraining the model from scratch with forget set data removed Bannihatti Kumar et al. (2023). However, given the enormous amount of resources required to retrain LLMs, it is generally more practical to employ *approximate unlearning* techniques that modify the behavior of a trained model in a post hoc manner. However, most previous LLM unlearning techniques require access to model internal weights Jang et al. (2023); Eldan & Russinovich (2023); Yao et al. (2023); Chen & Yang (2023); Meng et al. (2023); Wu et al. (2023), making them infeasible for black-box LLMs. For example, as two widely used unlearning algorithms, *Gradient Ascent* maximize the likelihood of forget set data, while *Data Relabeling* minimizes the likelihood of relabeled forget set data. Both of these methods require fine-tuning the LLMs. Black-box LLM unlearning is useful since this opens up the possibility of modular, customizable unlearning without the need to update the base LLM itself. Alternatively, in-context unlearning Pawelczyk et al. (2023) prompts LLMs with counterfactual forget set instances to steer model behavior at inference time. However, this approach comes with two major limitations. First, model developers still maintain an *explicit* list of sensitive information to be used during inference. Such practice is not only in violation of privacy regulations but also susceptible to malicious attacks such as prompting leaking Perez & Ribeiro (2022). Second, in-context unlearning cannot effectively deal with an ever-growing set of knowledge to be unlearned given the challenges of processing long context with LLMs Li et al. (2024). Tab. 1 summarizes the strengths and weaknesses of existing unlearning algorithms.

In this work, we propose $\delta$-UNLEARNING, an offset unlearning framework for arbitrary black-box LLM without updating its internal weights. Instead of tuning the black-box LLM itself, $\delta$-UNLEARNING learns the logit offset needed for unlearning by contrasting the logits from a pair of smaller, white-box models. During unlearning, we first compute the logit offset by taking the difference in logits from the two smaller models. Then, we add the logit offset between the two smaller models to the logits of the larger model. The intuition behind this is that we can learn the offset term that approximates how a larger model should modify its prediction in the face of sensitive queries from the behavior adaptation of a smaller model. $\delta$-UNLEARNING does not require access to the larger model's internal weights, nor retains any sensitive data for inference after unlearning. Our method also enables flexible version control and customization, since for different unlearning requests we only need to maintain a pool of smaller models, which can be combined with the same base LLM in a plug-and-play manner. This allows us to efficiently curate the pool of knowledge available to different applications using specialized unlearning modules, which is largely in line with previous efforts to modularize knowledge access for LLMs but from a different, complementary perspective Feng et al. (2024).

We evaluate the effectiveness of $\delta$-UNLEARNING on TOFU Maini et al. (2024), a widely used LLM unlearning benchmark containing knowledge about fictitious authors. Experiments show that when targeting the same forget performance, $\delta$-UNLEARNING maintains similar or even stronger performance on out-of-forget-scope data compared to directly fine-tuned larger models while requiring no parameter updates to the larger model.

Our contribution is three-fold. First, we propose $\delta$-UNLEARNING, an unlearning framework for arbitrary black-box LLM without modifying its parameters by only fine-tuning a smaller model to update the logits

of a larger one. Second, $\delta$-Unlearning can achieve the same level of unlearning as directly fine-tuning the larger model while still matching or even outperforming direct fine-tuning baselines on general tasks outside the unlearning scope. Third, $\delta$-Unlearning can be integrated into different unlearning algorithms, demonstrating the versatility of our approach.

## 2 Related Work

In this section, we summarize two lines of research that are highly related to our work.

**Machine Unlearning for LLM.** Prior works have explored machine unlearning as a way to mitigate the influence of undesirable training data on LLMs. Given the vast cost incurred by retraining LLMs from scratch Bannihatti Kumar et al. (2023), most unlearning methods apply post hoc finetuning or adaptation to steer the behavior on the forget set Jang et al. (2023); Eldan & Russinovich (2023); Yao et al. (2023); Chen & Yang (2023). Gradient ascent based methods fine-tune models by minimizing the likelihood of forget set data Jang et al. (2023); Chen & Yang (2023); Maini et al. (2024). Alternatively, several works proposed to maximize the likelihood of relabelled target data, where the original answer is replaced with a generic, insensitive response Eldan & Russinovich (2023); Patil et al. (2024). Auxiliary training objectives can also be introduced to maintain model performance on out-of-forget-scope data Yao et al. (2023); Wang et al. (2023). Another related line of research is model editing, where the goal is to identify and alter knowledge captured by local components within models Meng et al. (2023); Wu et al. (2023). While both model editing and unlearning attempt to modify the behavior of trained LMs, unlearning focuses on eliminating the effect of a specific set of training data without necessarily creating new answer mappings Liu et al. (2024c). It is worth noting that all of the aforementioned approaches require access to the model's internal weights. In-context unlearning Pawelczyk et al. (2023), while being applicable to black-box LLMs, still requires storing sensitive information for inference and therefore fails to address data privacy concerns. In this work, we propose an unlearning framework that does not require access to LLM weights, nor storage of sensitive information for inference.

**Logit Ensemble.** The potential of combining logits from different models has been studied in various context. One line of research focuses on controlling and improving LLM generation quality by contrasting the logits from different models or layers at decoding-time Liu et al. (2021); Shi et al. (2023); Li et al. (2023); Chuang et al. (2024). Logit ensemble has also been shown as an effective way of adapting LLMs to various downstream tasks. Ormazabal et al. (2023) propose to adapt LLMs to different domains through a learned combination with smaller domain experts. Mitchell et al. (2024) leverage an ensemble of difference-sized models to study the effect of pretraining and finetuning at different scales. Concurrently, Liu et al. (2024a) propose Proxy-Tuning that combines the logits from smaller tuned models with larger LLMs to enhance instruction following capabilities. Liu et al. (2024b) ensemble the logits of a main LLM with a paraphrase model that leads to a monotonic prompt paraphraser for rewriting prompts with enhanced generalizaion effects. Zhao et al. (2024) use the logits from unsafe LLMs to guide the jailbreaking of safer LLMs during decoding. In this work, we propose to utilize smaller LLMs to capture the logit offset needed for unlearning sensitive data from black-box LLMs while maintaining general performance on out-of-forget-scope tasks.

## 3 Method

In this section, we formulate the unlearning problem (§3.1), discuss the technical details of our $\delta$-Unlearning framework (§3.2), and highlight the strength of $\delta$-Unlearning compared to existing methods §3.3.

### 3.1 Problem Definition

Given a target forget set $S_f$ taken from the training data $S$ of an LLM $M$, the goal of unlearning is to obtain a new model $M'$ that resembles a model trained without $S_f$. This implies $M'$ should "forget" all information from the forget set without hurting the performance on out-of-forget-scope data. Ideally, unlearning can be accomplished by retraining $M$ on $S \backslash S_f$, i.e. the training set with forget set data removed. However, given

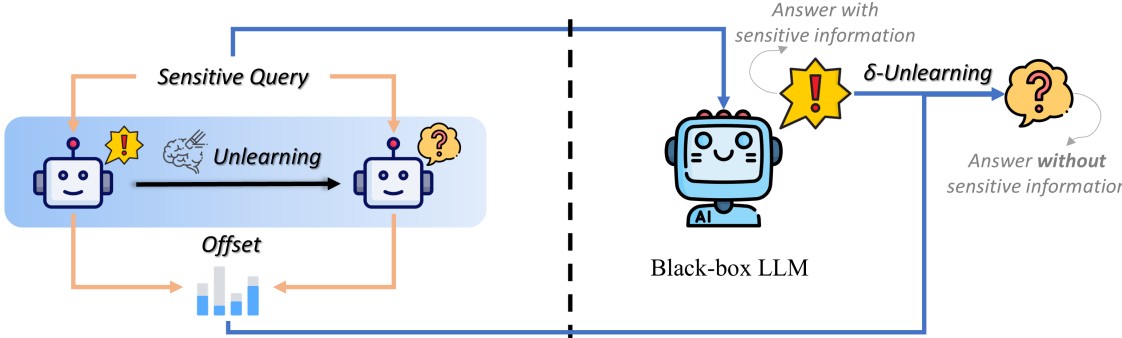

Figure 1: Overview of $\delta$-UNLEARNING. In order to adapt the behavior of a black-box LLM without updating its parameters, we combine it with a pair of smaller, white-box models (which we call offset models). For unlearning, we compute the logit offset of these two models and add it to the logits of the black-box LLM given the same query. Both of the two offset models are initialized from the same checkpoint, making the logit offset zero initially. The goal of $\delta$-UNLEARNING is to fine-tune one of them such that their logit offset, after being added to the logits of the black-box LLM, can steer its prediction away from generating sensitive information.

the prohibitive cost of retraining the LLM from scratch, it is generally more practical to approximate $M'$ by directly updating $M$. The unlearning problem can also optionally include a retain set $S_r$ on which the model after unlearning should not forget any information and maintain performance.

## 3.2 Offset Unlearning

$\delta$-UNLEARNING is based on the idea of a product-of-experts Hinton (2002) and its subsequent applications to ensemble of language models Liu et al. (2021); Meng et al. (2022); Li et al. (2023). Fig. 1 provides an overview of $\delta$-UNLEARNING.

Suppose we want to unlearn a forget set $S_f$ from an LLM $M$. Instead of directly updating the parameters of $M$, we introduce a pair of smaller, offset models $M_o$ and $M'_o$. We define their *logit offset* as the difference between the logits of two offset models $M'_o$ and $M_o$ given the same query. For unlearning, we add the logit offset to the logits of $M$ given the same query, essentially forming a logit ensemble of $M$, $M'_o$, and $M_o$. Both $M_o$ and $M'_o$ are initialized from the same checkpoint, making the logit offset zero for all data initially. During unlearning, we only update the parameters of $M'_o$ while keeping $M$ and $M_o$ frozen, and use the logit ensemble to generate the final output. In this way, we encourage $M'_o$ to deviate from its initialization $M_o$ given a sensitive query and learn the correct logit offset that applies to the logits of $M$, steering its prediction away from generating sensitive information. Formally, the logits of the ensemble $l_e$ are computed as follows:

$$l_e(y_t|q, y_{<t}) = l_M(y_t|q, y_{<t}) + \alpha(l_{M'_o}(y_t|q, y_{<t}) - l_{M_o}(y_t|q, y_{<t})),$$

where $l_M$, $l_{M'_o}$, and $l_{M_o}$ are the logits from their respective models, $q$ is the query, and $\alpha$ is a factor controlling the strength of applying the offset term to $M$. Since the logits are in the log space, the additive combination of them can also be interpreted as the following product-of-experts:

$$P_e(y_t|q, y_{<t}) \propto P_M(y_t|q, y_{<t}) \left( \frac{P_{M'_o}(y_t|q, y_{<t})}{P_{M_o}(y_t|q, y_{<t})} \right)^{\alpha}$$

Essentially, the probability of each token predicted by $M$ is scaled by the probability ratio between $M'_o$ and $M_o$, which reflects how $M'_o$ changes its token distribution relative to its initialization $M_o$ after unlearning. Specifically, when querying non-sensitive, out-of-forget-scope information, the probability ratio between $M'_o$ and $M_o$ should be close to one, making the token distribution of the ensemble similar to that of the original LLM $M$. When querying sensitive information that the model should forget, the token distribution of $M'_o$ differs from that of $M_o$ to adjust the probability ratio, thus steering the token distribution of the ensemble away from that of $M$.

During training, we optimize any unlearning objective on the prediction of the ensemble instead of on the original model $M$. For example, to unlearn the model using Gradient Ascent Jang et al. (2023); Chen & Yang (2023) where the objective is to minimize the likelihood of forget set data, we maximize the following loss function for instance $i$ of output length $l$:

$$\mathcal{L}_e^i = -\frac{1}{l} \sum_{t=1}^{l} \log P_e(y_t | q, y_{<t})$$

### 3.3 Merits of $\delta$-Unlearning

The design of $\delta$-UNLEARNING leads to the following key merits.

**Applicability to Black-box LLMs.** In contrast to most previous unlearning methods, $\delta$-UNLEARNING is applicable to not just open-sourced models, but also black-box LLMs without access to internal weights. Instead of updating $M$, $\delta$-UNLEARNING only obtains the logits from $M$, and learns the logit offset needed to adjust its prediction using smaller white-box models.

**Privacy Protection.** Prior work has proposed in-context unlearning Pawelczyk et al. (2023) to make unlearning possible for black-box LLMs. However, a key drawback of this approach is that the model developer still maintains an explicit, ever-growing list of sensitive information used to construct queries for unlearning during inference, which defeats the purpose of privacy protection. For comparison, $\delta$-UNLEARNING does not require storage of any explicit sensitive information after unlearning is completed.

**Version Control and Customization.** $\delta$-UNLEARNING also facilitates flexible version control and user customization, as instead of storing multiple versions of the larger model, we only need to keep track of a pool of smaller models. These models can be combined with the same base LLM in a plug-and-play manner. By using specialized unlearning modules, we can efficiently curate the pool of knowledge available to different applications.

## 4 Experiment

In this section, we provide a description of the evaluation setting (§4.1), a summary of baseline unlearning algorithms on which we apply our framework as well as other implementation details (§4.2), and the main results (§4.3).

### 4.1 Evaluation Setting

We conduct our experiments on TOFU Maini et al. (2024), a widely used unlearning benchmark designed for evaluating LLMs. The benchmark defines an unlearning task that targets information derived from a collection of fictitious author profiles that do not exist in real world. This creates a controlled unlearning setting with a well-defined unlearning scope and source of knowledge. TOFU designates a **Forget Set** which contains knowledge about a small subset of fictitious authors that we aim to unlearn. TOFU also includes three other datasets with knowledge about the retained fictitious authors (**Retain Set**), real world authors (**Real Author**), and general world facts (**World Fact**) respectively. Ideally, the model should retain all knowledge it had about these three retain datasets before and after unlearning.

The Forget Set evaluates *forget performance*, i.e., how well the model removes target information from its memory, while the latter three sets focus on *retain performance*, an indicator of how well the model maintains its performance on out-of-forget-scope data. The latter three sets also represent a series of out-of-forget-scope data with decreasing levels of relevance to the forget set. Generally speaking, it is more challenging for a model to remember out-of-forget-scope data that are more relevant to the forget set, a phenomenon known as knowledge entanglement Maini et al. (2024).

We follow the settings outlined in TOFU and report the following metrics for forget performance. **ROUGE** measures how well the generated output from the LLM matches the correct answer. Specifically, we use

the ROUGE-L recall score Lin (2004). **Probability** computes the conditional probability of the correct answer given the prompt. **Truth Ratio** measures how likely the correct answer is compared to a collection of wrong answers perturbed from the correct answer. Since the model is fine-tuned on one specific phrasing of the correct answer, thus potentially having inflated probability compared to other phrasing with similar meanings, Truth Ratio is computed using a paraphrased version of the original correct answer on the forget set and retain set. Following the original evaluation pipeline, we normalize Truth Ratio so that a higher truth ratio indicates better unlearning performance. **Forget Quality** measures the difference between distributions of Truth Ratios based on a Kolmogorov-Smirnov test. For retain performance, we report the aggregated **Model Utility**, which is computed by taking the harmonic mean of ROUGE-L, Probability, and Truth Ratio on all three retain datasets.

As we will demonstrate in §5.1, there is generally a trade-off between *forget performance* and *retain performance*. For example, a model can have a near-zero ROUGE score on the forget set but is completely unusable if the model always outputs gibberish given any prompt. Hence, we need to determine a target forget performance as a stopping criterion to facilitate direct comparison between different unlearning methods. In our experiments, we use the ROUGE score of the retraining baseline on the forget set as the stopping criterion, since retraining corresponds to an ideal scenario where the model has never been exposed to the forget set [1]. Following Yao et al. (2024), we match all models to the target ROUGE score by adjusting the learning rate.

In addition to TOFU, we assess if the unlearned model preserves general utilities on well-established benchmarks, including ARC Clark et al. (2018), HellaSwag Zellers et al. (2019), WinoGrande Sakaguchi et al. (2021) and OpenBookQA Mihaylov et al. (2018). Since solving these general tasks does not involve knowledge about the data we aim to unlearn, the model after unlearning should maintain as much performance as possible. We follow the default evaluation setting from Gao et al. (2023) and report accuracy on all four tasks under the zero-shot setting.

### 4.2 Model Configuration

**Unlearning Algorithms.** $\delta$-UNLEARNING is a general unlearning framework compatible with different existing unlearning algorithms. We compare $\delta$-UNLEARNING with its corresponding direct fine-tuning baseline when incorporated with each of the following commonly used unlearning algorithms. *Gradient Ascent* Jang et al. (2023); Chen & Yang (2023) minimizes the likelihood of the forget set. *Gradient Difference* Liu et al. (2022); Yao et al. (2023) minimize forget set likelihood while maximize retain set likelihood. *KL Minimization* Maini et al. (2024) penalizes the distributional distance between models before and after unlearning. *Data Relabeling* Eldan & Russinovich (2023) trains the model on forget set questions paired with an alternative answer that abstains from answering the question such as "I don't have that information." We also include the *Retraining* baseline which fine-tunes the initial model with the forget set excluded, which serves as the upper bound in terms of balancing forget and retain performance .

**Implementation.** We run our experiments on the widely used Llama2 model family Touvron et al. (2023). Specifically, we use *Llama2-13b-chat-hf* as the larger model and *Llama2-7b-chat-hf* as the smaller offset model. Note that while Llama2 models All models are trained using NVIDIA A100 GPUs for 5 epochs with a batch size of 32. We set $\alpha$ to 1 for our experiments.

### 4.3 Main Results

Our experimental results on TOFU are shown in Tab. 2. The model before unlearning exhibits strong memorization over both the forget set and retain set, indicated by high ROUGE and probability scores. This is as expected since the model is explicitly trained on the full dataset of fictitious authors to simulate model's exposure to private information. Retraining significantly reduces the model's knowledge on the forget set while maintaining similar model utility as it is before unlearning. Although retraining would not

---

[1] By definition the retraining baseline has a forget quality of 1.0, representing an upper bound for this metric. Hence, it is infeasible to use forget quality as the stopping criterion for forget performance.

| Method | TOFU Forget | | | | TOFU Retain | ARC | HS | WG | OBQA |
|---|---|---|---|---|---|---|---|---|---|
| | RL (↓) | P (↓) | TR (↑) | FQ (↑) | MU (↑) | Acc | Acc | Acc | Acc |
| *Before Unlearning* | 95.6 | 98.3 | 49.5 | 1.3e-13 | 62.1 | 44.0 | 58.1 | 67.9 | 36.2 |
| *Retraining* | 38.9 | 15.2 | 65.6 | 1.0 | 62.9 | 45.0 | 58.5 | 68.0 | 34.6 |
| *Gradient Ascent* | | | | | | | | | |
| Direct Fine-tuning | 38.8 | 3.4 | 53.3 | 2.6e-7 | 32.7 | 39.9 | 56.4 | 65.2 | 34.4 |
| δ-Unlearning | 38.6 | 15.2 | 57.9 | 4.0e-6 | 48.6 | 42.2 | 56.3 | 65.7 | 32.8 |
| *Gradient Difference* | | | | | | | | | |
| Direct Fine-tuning | 38.9 | 2.1 | 51.9 | 1.4e-6 | 51.4 | 40.4 | 56.3 | 64.9 | 32.6 |
| δ-Unlearning | 38.1 | 6.2 | 52.5 | 6.7e-6 | 50.5 | 40.9 | 55.7 | 65.2 | 35.4 |
| *KL Minimization* | | | | | | | | | |
| Direct Fine-tuning | 39.8 | 3.1 | 53.4 | 1.4e-6 | 33.5 | 39.2 | 56.5 | 65.0 | 34.0 |
| δ-Unlearning | 39.6 | 14.1 | 57.5 | 1.8e-5 | 50.4 | 43.7 | 57.2 | 66.9 | 34.4 |
| *Data Relabeling* | | | | | | | | | |
| Direct Fine-tuning | 38.1 | 92.5 | 53.3 | 6.6e-12 | 56.1 | 43.5 | 57.9 | 68.9 | 34.6 |
| δ-Unlearning | 36.3 | 91.5 | 50.8 | 3.0e-13 | 58.6 | 44.2 | 58.0 | 68.0 | 34.8 |

Table 2: Results on TOFU and general benchmarks. We report ROUGE-L recall (RL), Probability (P), Truth Ratio (TR) and Forget Quality (FQ) on the Forget Set and Model Utility (MU) on retain data from the TOFU benchmark. We report accuracy on general benchmarks. Higher scores are better except ROUGE and probability on the Forget Set. Better scores are underlined for each of the four unlearning strategies.

| Method | Retain Set | | | Real Author | | | World Fact | | |
|---|---|---|---|---|---|---|---|---|---|
| | RL | P | TR | RL | P | TR | RL | P | TR |
| *Before Unlearning* | 96.3 | 97.9 | 51.2 | 85.2 | 44.5 | 55.7 | 87.7 | 42.5 | 56.3 |
| *Retraining* | 95.8 | 97.7 | 50.4 | 89.5 | 45.8 | 58.5 | 85.5 | 43.0 | 57.4 |
| *Gradient Ascent* | | | | | | | | | |
| Direct Fine-tuning | 51.2 | 8.0 | 51.6 | 52.3 | 43.9 | 58.3 | 80.2 | 44.6 | 60.6 |
| δ-Unlearning | 41.0 | 26.1 | 48.9 | 75.0 | 45.3 | 57.4 | 82.1 | 47.0 | 63.7 |
| *Gradient Difference* | | | | | | | | | |
| Direct Fine-tuning | 56.8 | 58.9 | 55.1 | 61.4 | 35.0 | 47.9 | 80.4 | 38.9 | 53.7 |
| δ-Unlearning | 53.4 | 47.8 | 51.9 | 60.6 | 36.1 | 45.9 | 83.2 | 41.3 | 59.1 |
| *KL Minimization* | | | | | | | | | |
| Direct Fine-tuning | 53.0 | 8.4 | 51.0 | 55.8 | 42.2 | 56.4 | 83.3 | 43.3 | 58.8 |
| δ-Unlearning | 46.1 | 27.9 | 50.9 | 80.4 | 45.1 | 57.5 | 84.9 | 46.3 | 64.0 |
| *Data Relabeling* | | | | | | | | | |
| Direct Fine-tuning | 85.0 | 95.3 | 48.0 | 82.5 | 38.0 | 46.3 | 87.7 | 39.2 | 49.2 |
| δ-Unlearning | 72.4 | 95.1 | 49.6 | 78.7 | 41.5 | 52.6 | 86.9 | 42.3 | 55.5 |

Table 3: Detailed results on TOFU retain data, namely the Retain Set, Real Author and World Fact. Higher numbers are better for all metrics.

be feasible in real world scenarios, its performance gives us a better understanding of the gap between exact unlearning and post hoc approximate unlearning methods.

We first examine the forget quality of different post-hoc unlearning methods on the TOFU Forget Set. As shown in Tab. 2, both direct fine-tuning and δ-Unlearning can reach a level of unlearning similar to

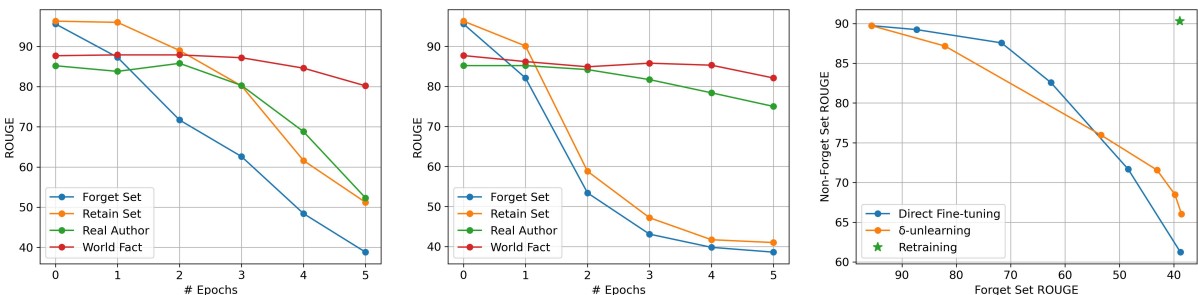

Figure 2: Unlearning trajectory of Gradient Ascent using direct fine-tuning (left), $\delta$-UNLEARNING (middle), and the tradeoff curve between forget and retain performance (right) over the course of unlearning. For training trajectories we report ROUGE score on all four TOFU datasets. For the tradeoff curve we report Forget Set ROUGE versus Non-forget Set ROUGE score.

retraining in terms of ROUGE score of the generated response. Although direct fine-tuning tends to assign lower probabilities to the correct answer on 3 out of the 4 methods we investigate, $\delta$-UNLEARNING produces a higher truth ratio and forget quality in all three cases. A higher truth ratio is desirable since it indicates the presence of other highly likely alternatives, making the correct answer less distinguishable from other wrong answers.

Interestingly, data relabeling retains a very high probability score and very low forget quality despite having a similarly low ROUGE score as other algorithms on the forget set. This is likely due to relabeling being the only method that does not explicitly minimize the likelihood of the original forget set answers.

We then investigate how well the unlearned model maintains its performance on data outside the unlearning scope. On TOFU retain data, $\delta$-UNLEARNING preserves more model utility than direct fine-tuning on 3 out of the 4 methods we compare. In particular, $\delta$-UNLEARNING demonstrates superior retain performance when applying to gradient ascent and KL minimization, beating direct fine-tuning by more than 15 points. Taking a closer look at how these models perform on individual TOFU retain data, we observe in Tab. 3 that direct fine-tuning tends to perform better on the Retain Set, while $\delta$-UNLEARNING outperforms direct fine-tuning on the Real Author and World Fact. This indicates a slight divergence between direct fine-tuning and $\delta$-UNLEARNING in terms of how to balance performance across different types of knowledge during unlearning. This is likely a result of changing training dynamics with the introduction of offset models, which we will study in more detail in §5.1

In addition to TOFU, we also evaluate the utility of the unlearned model on general task benchmarks. Performance on these tasks is also an important indicator of retain performance, reflecting whether general capabilities of LLMs are preserved after unlearning. As shown in Tab. 2, $\delta$-UNLEARNING achieves competitive performance on most metrics when compared to direct fine-tuning baselines, and closes the performance gap between before and after unlearning. $\delta$-UNLEARNING consistently outperforms direct fine-tuning with KL minimization, and bring improvement on 3 out of 4 tasks with gradient difference and data relabeling.

Overall, our experiments demonstrate that $\delta$-UNLEARNING is a strong alternative to direct fine-tuning, with matching or even superior performance in terms of both forget and retain performance. $\delta$-UNLEARNING is also effective across different unlearning algorithms, showing the versatility of our approach.

## 5 Analysis

In this section, we provide analyses on the training trajectory of the unlearning process (§5.1), and the effect of varying offset strength (§5.2).

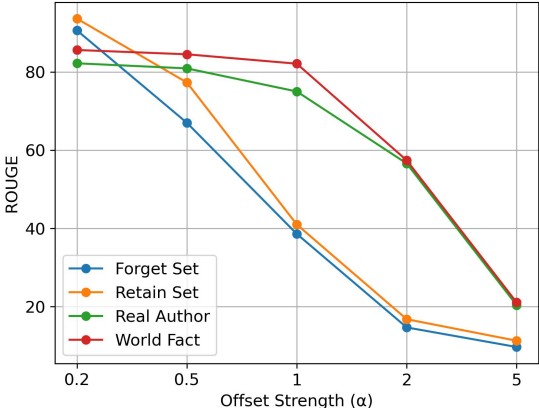

Figure 3: Effect of varying offset strength on the model after $\delta$-Unlearning with Gradient Ascent.

## 5.1 Unlearning Trajectory

To better understand how forget and retain performance change over the course of unlearning, we study the training trajectory of both direct fine-tuning and $\delta$-Unlearning. As shown in Fig. 2, when targeting the same ROUGE score on the forget set, $\delta$-Unlearning exhibits a steeper decline on the forget set initially compared to direct fine-tuning, which is also coupled with a steeper decline on the Retain Set. As unlearning progresses, direct fine-tuning starts to lose performance on the Real Author set that the model should not forget, while $\delta$-Unlearning still maintains relatively stable performance.

When comparing the training trajectory on different TOFU datasets, we can clearly observe the varying degrees of knowledge entanglement with the Forget Set. Being the most similar to the Forget Set, Retain Set performance starts to degrade at early stages, followed by the Real Author set. Performance on the World Fact, which is the least relevant to the Forget Set, only declines slightly towards the end of unlearning. This highlights the importance of finding a good balance between forget and retain performance for an unlearning method. We also study this trade-off from a more direct perspective by plotting the curve forget set ROUGE score versus non-forget set ROUGE score in Fig. 2 (right). A desired model should lie at the upper right corner, which represents strong forget and retain performance. While Direct fine-tuning maintains more performance on retain data initially, $\delta$-Unlearning achieves a better balance at higher unlearning levels as direct fine-tuning starts to lose more performance on non-forget sets.

## 5.2 Effect of Offset Strength

As we mentioned in §3.2, we can adjust the value of $\alpha$ to control the strength of the logit offset being added to the larger model's logits. We experiment with using difference $\alpha$ values during inference and study its effect on forget and retain performance. As shown in Fig. 3. A low offset strength makes the effect of logit offset negligible, and the prediction of the ensemble is essentially dominated by the larger model $M$ without unlearning. As we gradually increase the offset strength, the unlearning effect becomes more prominent and forget set ROUGE score decreases significantly. Similar to what we observe in Fig. 2, Retain Set performance largely follows the trajectory of Forget Set, while Real Author and World Fact performance are less influenced by the increase of unlearning offset strength. When we surpass the level of offset strength used in training, we observe continued performance degradation on all four datasets. The ROUGE score on forget set drops below 10 when $\alpha$ increases to 5, a score much lower than the retraining baseline (which has a ROUGE score of 38.9). However, at this offset strength the model becomes unusable, indicated by poor performance across all three non-forget set.

We present an example from the Forget Set in Tab. 4 to provide a better understanding of model behavior at different offset strength levels. At $\alpha$=0.2, the model can perfectly reproduce the answer that is supposed

|  | Example |
|---|---|
| Sensitive Query | In which genre does Hina Ameen primarily write? |
| Ground Truth | Hina Ameen primarily contributes to the geology genre. |
| $\alpha = 0.2$ | Hina Ameen primarily contributes to the geology genre. |
| $\alpha = 0.5$ | Hina Ameen primarily contributes to the genre of geology. Her extensive knowledge of ... |
| $\alpha = 1.0$ | Hina Ameen works primarily in the genre of mythology. Her literature has a deep connection ... |
| $\alpha = 2.0$ | As the book primarily consist narrations revolved historical Daker period ... |
| $\alpha = 5.0$ | As writers focus deep introsvosity embits poert calurity reveiased world literature reflect ... |

Table 4: Example response by $\delta$-UNLEARNING on the Forget Set with varying offset strength during inference.

| Model | TOFU Forget | | | | TOFU Retain |
|---|---|---|---|---|---|
| | RL ($\downarrow$) | P ($\downarrow$) | TR ($\uparrow$) | FQ ($\uparrow$) | MU ($\uparrow$) |
| Retraining | 37.2 | 11.6 | 57.7 | 1.0 | 58.8 |
| 7B Direct Fine-tuning | 37.4 | 4.1 | 50.1 | 1.2e-4 | 37.8 |
| 7B + 3B Offset | 37.3 | 37.0 | 46.7 | 1.5e-7 | 52.2 |
| 7B + 1.5B Offset | 36.9 | 17.5 | 49.9 | 6.7e-6 | 49.1 |
| 7B + 0.5B Offset | 37.0 | 14.7 | 43.8 | 4.9e-10 | 48.8 |

Table 5: Results on different offset model size.

to be forgotten, showing that unlearning is not taking effect at low offset strength. At $\alpha$=0.5, the model is still capable of recalling the correct answer, despite in slightly different phrasing. At $\alpha$=1, we obtain a fluent response with the sensitive information from the ground truth removed, demonstrating success of unlearning. If we further increase offset strength, the model starts to generate gibberish and eventually becomes unusable. In conclusion, using the same offset strength as in training leads to the best results overall.

## 5.3 Choice of Offset Model

To study how the choice of offset model affects unlearning performance, we run a series of experiments using Qwen family models as they offer a wide range of model sizes. Specifically, we apply offset unlearning with gradient ascent to a target Qwen2.5-7b-instruct model using 3B, 1.5B and 0.5B offset models. As shown in Tab. 5, all models can reach a level of unlearning similar to retraining as measured by ROUGE score of the generated responses, and using a stronger offset model generally leads to better performance on retain sets. Distributional metrics such as truth ratio and forget quality are more sensitive to the choice of offset model especially when the offset models are weak, as weak offset models tend to behave very differently from the target model during unlearning, and thus making the final logit ensemble deviate more from direct fine-tuning. Similar to what we observe in Tab. 3, using different offset models affects how the model balances performance across different types of knowledge during unlearning.

## 6 Conclusion

In this work, we propose $\delta$-UNLEARNING, an offset unlearning framework applicable to black-box LLM that does not require access to model's internal weights. Instead of modifying model parameters, $\delta$-UNLEARNING learns the logit offset needed to steer model behavior on the target forget set data. Experiments show that $\delta$-UNLEARNING is on par with and sometimes even stronger than direct fine-tuning in terms of both forget quality and model utility. We also demonstrate that $\delta$-UNLEARNING is compatible and effective when combined with various unlearning algorithms, thus providing a versatile solution to adapting existing algorithms to black-box LLMs.

## Acknowledgment

We appreciate the reviewers for their insightful comments and suggestions. James Y. Huang was supported by a gift fund from the USC Center on Secure & Trusted ML. Fei Wang was supported by an Amazon ML PhD Fellowship. Muhao Chen was supported by the DARPA FoundSci Grant HR00112490370, the NSF of the United States Grant ITE 2333736.

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
