# OpenReview forum: "Offset Unlearning for Large Language Models"
_TMLR — Accepted by TMLR_

### Review · Reviewer_wcJf · 2025-02-25

**Summary Of Contributions:**

1. This paper summarizes the drawbacks of current unlearning methods for large language models (LLMs) and proposes a logit offset method to reduce reliance on model weights, addressing the challenges posed by the ever-growing forgetting set.

2. Experimental results demonstrate that the proposed method achieves performance comparable to direct fine-tuning approaches on LLMs.

**Audience:**

Yes

**Broader Impact Concerns:**

No further concerns

**Claims And Evidence:**

No

**Requested Changes:**

1. This paper can provide more details about how to select the smaller models and whether different smaller models can have effects on the larger model.

2. This paper can provide more explanation about how the proposed method can solve the two drawbacks of current LLM unlearning methods

3. Then, regarding the experiment results, this paper can provide a more comprehensive result to prove the proposed method can be superior to other methods

**Strengths And Weaknesses:**

Strengths:

1. The problem of unlearning in large language models (LLMs) is critical, and the introduction effectively presents the issue with clarity.

2. This paper highlights common drawbacks of existing LLM unlearning methods and proposes a systematic approach that can be integrated with other unlearning techniques.

3. The experimental results demonstrate the efficiency of the proposed method across various unlearning tasks.

Weaknesses:

1. Some parts of the methodology are unclear—for instance, the process for selecting the smaller model $M_o$ is not well-defined.

2. Although the paper claims that previous unlearning methods require an explicit list of sensitive information, the model offset stage still depends on forgetting data for unlearning small models. Additionally, it does not address the growing size of the forgetting data, an issue that is omitted from both the methodology and experimental sections.

3. The experimental results are not entirely convincing, as direct fine-tuning outperforms the proposed method on several evaluations.

4. The experiments only compare the proposed method with direct fine-tuning without providing a detailed explanation of the fine-tuning process, nor do they explore a broader range of small models.

---

> ### Author Response · Authors · 2025-03-22
>
> We greatly appreciate your feedback. Below are our responses to your questions.
>
> **Offset model selection**
>
> $\delta$-unlearning can be applied as long as the offset model has a matching token vocabulary with the target model so that the logit ensemble can be computed. We have added experiments in Section 5.3 to study how the choice of offset model affects unlearning performance. Using a bigger offset model tends to give stronger retain performance at similar levels of unlearning on forget set as measured by ROUGE score of generated response.
>
>
> **Drawbacks of existing methods**
>
> Previous black-box unlearning methods such as in-context unlearning require sensitive information to be stored explicitly and indefinitely for inference. This is not the case for finetuning-based methods since the forgetting data is no longer needed once the model has unlearned the data, and new forgetting data can be continually unlearned. Our method essentially brings the benefits of finetuning-based unlearning to black-box models with no weight access.
>
>
> **Experimental results**
>
> As we discussed in Section 4.2, our experiments focus on comparing two paradigms of unlearning:
> * Direct fine-tuning: we assume accessible target model weights and unlearning is performed by directly finetuning the target model itself.
> * Offset unlearning: we assume no access to target model weights and unlearning is performed by finetuning an external small model to learn a logit offset.
>
> We demonstrate that for four existing unlearning algorithms that were previously implemented in the form of direct fine-tuning, we can adapt them into their offset unlearning counterparts, and achieve comparable performance without finetuning the target model itself (which is a significant restriction we impose to facilitate black-box unlearning).

---

### Review · Reviewer_2dTb · 2025-03-06

**Summary Of Contributions:**

The paper introduces a novel framework for unlearning sensitive or problematic information from large language models without accessing their internal weights. Instead of fine-tuning the large, black-box model directly, the method learns a logit offset using a pair of smaller, white-box offset models. This approach is designed to modify the output distribution of the target model while preserving its overall performance on non-sensitive tasks. The framework is versatile, as it can be combined with various existing unlearning algorithms, and it demonstrates empirical effectiveness on a synthetic benchmark (TOFU) as well as on several general task benchmarks.

**Audience:**

Yes

**Broader Impact Concerns:**

The proposed framework, if misused, could potentially erase information deliberately for unethical purposes. Outlining safeguards, transparency measures, or audit mechanisms for the unlearning process could mitigate the risk of misuse and promote responsible deployment.

**Claims And Evidence:**

No

**Requested Changes:**

1. It's important to include empirical studies that examine the effect of using offset models with different architectures from the target LLM. In addition, it would strongly support the claim of effective black-box unlearning by demonstrating robustness when the offset model diverges in structure, size, or training protocol from the large model.

2. The authors should discuss and quantify the additional computational overhead introduced by maintaining an ensemble of models. It would be informative to also have a trade-off analysis showing how the size of the offset models relates to unlearning performance (and whether smaller models might suffice compared to larger ones).

3. Since the results are sensitive to the offset strength $\alpha$, it would be helpful to discuss strategies for selecting the optimal $\alpha$ or an adaptive mechanism for setting $\alpha$.

4. I suggest including the ROUGE score of the retraining baseline in Fig. 2, as this would help understand how δ-Unlearning performs when unlearning levels are near those achieved by full retraining, rather than only at higher unlearning levels.

**Strengths And Weaknesses:**

1. The approach is conceptually innovative by leveraging logit offsets to modify a black-box model’s outputs. However, the experiments rely solely on closely related llama2 variants. This raises concerns about whether the mechanism holds when the offset model’s architecture significantly differs from that of the large model. Moreover, even when using similar architectures, factors like different initialization and training recipes might yield different logits, so the claim of general “black-box unlearning” appears a bit overextended.

2. The framework’s ability to integrate with multiple unlearning algorithms is a strength, as is its potential for preserving privacy by not storing sensitive information at inference time. On the downside, maintaining a pool of smaller offset models introduces system complexity and extra overhead, and it is unclear how mismatches in architecture or size between the large model and offset models might affect overall performance. This additional layer might complicate real-world deployment and scalability.

3. Empirical evaluations on the TOFU benchmark and general tasks show promising trade-offs between forgetting and retaining knowledge. However, the results seem to be sensitive to the offset strength parameter $\alpha$. Without a robust strategy for tuning $\alpha$, practical deployment might be challenging, especially if the optimal value is difficult to determine and maintain across different scenarios.

---

> ### Author Response · Authors · 2025-03-22
>
> We greatly appreciate your feedback. Below are our responses to your questions.
>
> **Offset model selection and computational overhead**
>
> $\delta$-unlearning can be applied as long as the offset model has a matching token vocabulary with the target model so that the logit ensemble can be computed. We have added experiments in Section 5.3 to study how the choice of offset model affects unlearning performance. Using a bigger offset model tends to give stronger retain performance at similar levels of unlearning on forget set as measured by ROUGE score of generated response. In the meantime, using a smaller offset model reduces computational overhead. For example, compared to directly fine-tuning a 7B target model, using 3B and 0.5B offset models leads to a x1.7 and x1.4 slowdown during inference respectively. The current implementation of offset unlearning serves as a proof of concept and still has significant room for runtime optimization, such as parallelizing inference of multiple models.
>
> **Sensitivity to offset strength**
>
> While we experiment with varying the offset strength during inference in section 5.2, it is recommended to always use the same offset strength as in training to achieve optimal performance (in our experiments we used $\alpha$=1 as mentioned in section 4.2). Intuitively this makes sense because the logit offset is learned with respect to the \alpha value used during training. In Table 4 we demonstrate that setting $\alpha$=1 indeed gives the most fluent response with sensitive information removed. We have revised section 5.2 to clarify this.
>
> **Including retraining baseline in Figure 2**
>
> We have updated Figure 2 to include the retraining baseline. Since retraining is not an unlearning algorithm, there is no unlearning trajectory for retraining. Thus the performance of retraining is represented as a single point at the upper right in Figure 2 (right).

---

### Review · Reviewer_d6iE · 2025-03-06

**Summary Of Contributions:**

The paper introduces **δ-Unlearning**, a novel offset unlearning framework for **black-box Large Language Models (LLMs)** that allows unlearning specific training data without modifying the model’s internal weights. The key contributions are:

1. **Introduction of δ-Unlearning**: The proposed method enables unlearning in black-box LLMs by leveraging a **logit offset** technique. Instead of fine-tuning the LLM itself, **δ-Unlearning computes and applies a logit offset using two smaller, white-box models** that contrast the logits of sensitive versus general knowledge.

2. **Effective Unlearning While Preserving General Performance**: Experiments show that δ-Unlearning achieves **comparable or superior performance** to traditional fine-tuning methods in removing targeted knowledge while maintaining accuracy on tasks outside the forget scope.

3. **Versatility Across Unlearning Methods**: δ-Unlearning is **compatible with various unlearning algorithms** such as **Gradient Ascent, Gradient Difference, KL Minimization, and Data Relabeling**, making it a **flexible** and **modular** approach.

4. **Black-Box Applicability and Privacy Compliance**: Unlike previous methods that require access to LLM **weights** or store sensitive data at inference, δ-Unlearning can **unlearn data from black-box models without violating data protection regulations**.

5. **Efficient Version Control and Adaptability**: The approach **modularizes unlearning**, enabling users to **curate knowledge** by maintaining a pool of smaller models that can be combined in a **plug-and-play** manner with the base LLM.

6. **Extensive Empirical Validation**: The method is evaluated on the **TOFU benchmark**, showing strong unlearning capabilities while maintaining general model utility. It also demonstrates robust performance on general NLP benchmarks, proving its adaptability beyond targeted unlearning tasks.

### **New Knowledge Introduced**
- **Offset-based unlearning as an alternative to model retraining**: δ-Unlearning shows that modifying **only the logits** (rather than model weights) can achieve effective unlearning.
- **Modular and interpretable unlearning for black-box LLMs**: The approach allows for **targeted and customizable** knowledge removal without violating **privacy constraints**.
- **Trade-offs between forgetting and retaining knowledge**: The paper provides **insightful analysis** on how different **unlearning strengths** impact knowledge retention and general model performance.

**Audience:**

Yes

**Claims And Evidence:**

Yes

**Requested Changes:**

### **Critical Changes (Necessary for Acceptance)**

1. **Expand Real-World Evaluation Beyond TOFU**
   - While TOFU is a useful benchmark, the paper would significantly **benefit from experiments on real-world unlearning scenarios**, such as:
     - Removing **personally identifiable information (PII)** from LLM outputs.
     - Forgetting **copyrighted or proprietary content**.
     - Addressing **biased or harmful responses**.
   - This would strengthen the claim that **δ-Unlearning is practical for real-world applications**.

2. **Analyze the Impact of Offset Model Quality**
   - δ-Unlearning relies on **two smaller white-box models** to generate logit offsets. The paper should **investigate how the choice of these models affects unlearning performance**.
   - Key questions to explore:
     - Does the size of the **offset models** influence the effectiveness of unlearning?
     - How do different training strategies for these smaller models affect the **final logit offsets**?
     - Are there cases where δ-Unlearning fails due to poor offset model choices?
   - If these factors significantly impact the method's effectiveness, **guidelines should be provided** for selecting appropriate offset models.

3. **Provide a More Detailed Analysis of Forget vs. Retain Trade-offs**
   - The paper currently discusses the **trade-off between unlearning and retention**, but further **quantitative insights** would be useful.
   - Suggested improvements:
     - Show how different **types of knowledge (e.g., factual, contextual, reasoning-based) are affected** by δ-Unlearning.
     - Provide **more granular analysis** on how performance degrades across different datasets in TOFU.
     - Discuss whether **δ-Unlearning disproportionately affects certain types of information**, leading to unintended consequences.

4. **Clarify Computational Efficiency and Overhead**
   - While δ-Unlearning avoids full model retraining, it **introduces additional computational steps** through offset model training.
   - The paper should include:
     - A direct **comparison of computational cost** (e.g., FLOPs, memory usage, or wall-clock time) between δ-Unlearning and traditional fine-tuning-based unlearning.
     - Discussion of whether δ-Unlearning scales effectively for **larger LLMs** (e.g., 30B+ parameters).

### **Recommended Changes (Would Strengthen the Paper)**

5. **Improve the Explanation of Logit Offsets and Their Impact**
   - The paper presents the mathematical formulation of **logit offsets**, but some parts could be more **intuitively explained**.
   - Suggested improvements:
     - **Add a visualization** showing how **logits shift before and after unlearning**.
     - Provide **concrete examples** where δ-Unlearning successfully **removes unwanted knowledge**.
     - Offer a **more detailed interpretation** of how the logit offset technique works in practice.

6. **Discuss Adaptive or Automated Offset Strength Selection (α)**
   - The paper currently sets **α = 1**, but there is **no discussion on how to tune α optimally**.
   - Suggested additions:
     - Investigate **different ways to select α dynamically** instead of fixing it.
     - Explore whether an **adaptive α** can improve unlearning effectiveness while minimizing unintended knowledge loss.

7. **Broaden the Related Work Section**
   - The discussion on **machine unlearning** is well-grounded, but adding references to **recent advances in continual learning, domain adaptation, and knowledge editing** could provide **additional context**.

8. **Enhance Paper Readability and Structure**
   - Some explanations, particularly in **Section 3 (Method) and Section 5 (Analysis)**, could be **streamlined** for better readability.
   - Specific improvements:
     - Reword complex mathematical passages for clarity.
     - Make the **figures and tables more self-explanatory** by adding **more detailed captions**.
     - Reduce **redundant phrasing** in discussions of unlearning benchmarks.

**Strengths And Weaknesses:**

#### **Strengths**

1. **Novel Unlearning Framework for Black-Box LLMs**
   - The introduction of **δ-Unlearning** offers a **practical, parameter-free** solution for unlearning in **black-box LLMs**.
   - Unlike existing methods, it does not require **fine-tuning the LLM itself**, making it **cost-effective** and **applicable to proprietary models**.

2. **Versatility Across Unlearning Algorithms**
   - δ-Unlearning is designed to be **modular** and **integrates well** with different unlearning techniques (**Gradient Ascent, Gradient Difference, KL Minimization, and Data Relabeling**).
   - This flexibility increases the method's **applicability across different use cases**.

3. **Stronger Privacy Compliance**
   - The method does not **store sensitive data** or **require access to model weights**, addressing concerns raised by **data protection laws** like the GDPR.
   - Avoids **in-context unlearning limitations**, where storing sensitive information for inference can **violate privacy**.

4. **Improved Retain Performance Compared to Direct Fine-Tuning**
   - **Maintains or even enhances performance** on out-of-forget-scope tasks compared to standard fine-tuning approaches.
   - Empirical results on **TOFU and general NLP benchmarks** (ARC, HellaSwag, WinoGrande, OpenBookQA) **demonstrate its ability to preserve general knowledge**.

5. **Scalability and Version Control**
   - The approach **enables efficient version control** by maintaining **a pool of smaller models** instead of retraining the entire LLM.
   - Supports **customizable and modular unlearning**, allowing **domain-specific** knowledge updates without affecting the base LLM.

6. **Thorough Experimental Validation**
   - The **TOFU benchmark** is well-suited for evaluating unlearning, providing a **controlled setting** with fictitious author knowledge.
   - Evaluation metrics (ROUGE, Probability, Truth Ratio, Forget Quality) give a **clear assessment of both forgetting and retention trade-offs**.

---

#### **Weaknesses / Areas for Improvement**

1. **Dependency on Smaller White-Box Models**
   - **δ-Unlearning relies on training smaller white-box models** to compute logit offsets, which may **limit its applicability** in environments where **such models are unavailable** or where creating high-quality smaller models is challenging.
   - The effectiveness of the approach may **depend on the quality of the smaller models**, which is not fully analyzed in the paper.

2. **Lack of Real-World Evaluation Beyond TOFU**
   - While TOFU is a **strong benchmark**, the paper does not test δ-Unlearning on **real-world unlearning tasks**, such as removing **sensitive, copyrighted, or biased content** from an LLM.
   - Evaluating on datasets with **legal or ethical constraints** (e.g., **removing personally identifiable information**) would further validate the framework.

3. **Limited Analysis of Trade-Offs in Forget and Retain Performance**
   - Although the paper discusses the **trade-offs between forgetting and retention**, further **fine-grained analysis** of when **forgetting causes unintended knowledge loss** (knowledge entanglement) would strengthen the claims.
   - A **more detailed breakdown** of how δ-Unlearning affects different types of knowledge (e.g., factual vs. contextual) would be useful.

4. **Hyperparameter Sensitivity of Logit Offsets (α)**
   - The offset strength parameter **α** is crucial for effective unlearning, yet **there is limited discussion** on how to **optimally select α**.
   - Further experiments on **adaptive or automated tuning** of α based on model behavior could enhance applicability.

5. **Computational Overhead**
   - While δ-Unlearning **avoids full model retraining**, training **additional offset models** adds some **computational cost**.
   - A direct **comparison of computational efficiency** with traditional unlearning methods (in terms of FLOPs or training time) would be beneficial.

6. **Clarity in Presentation and Explanation of Some Concepts**
   - The **mathematical formulation** of logit offsets and their influence on LLM predictions could be more **intuitively explained** for a broader audience.
   - A **visualization of how logits shift** before and after unlearning would provide better clarity.

---

> ### Author Response · Authors · 2025-03-24
>
> We greatly appreciate your feedback. Below are our responses to your questions.
>
> **Offset model selection and computational overhead**
>
> We have added experiments in Section 5.3 to study how the choice of offset model affects unlearning performance. Using a bigger offset model tends to give stronger retain performance at similar levels of unlearning on forget set as measured by ROUGE score of generated response. In the meantime, using a smaller offset model reduces computational overhead. For example, compared to directly fine-tuning a 7B target model, using 3B and 0.5B offset models leads to a x1.7 and x1.4 slowdown during inference respectively. The current implementation of offset unlearning still has room for optimization, such as parallelizing inference of multiple models.
>
> **Detailed trade-off analysis**
>
> In Table 2 we already demonstrate that direct finetuning and $\delta$-unlearning have similar performance across four general domain datasets covering science, common sense knowledge and reasoning. In addition, we have added a breakdown and analysis of model performance on different TOFU subsets in Section 4.3 and Table 3.
>
> **Offset strength setting**
>
> While we experiment with varying the offset strength during inference in section 5.2, it is recommended to always use the same offset strength as in training to achieve optimal performance (in our experiments we used $\alpha$=1 as mentioned in section 4.2). Intuitively this makes sense because the logit offset is learned with respect to the \alpha value used during training. In Table 4 we demonstrate that setting $\alpha$=1 indeed gives the most fluent response with sensitive information removed. We have revised section 5.2 to clarify this.

---

### Decision · Action_Editor_8MMg · 2025-04-18

**Recommendation:** Accept as is

**Comment:**

The paper introduces a novel framework for unlearning sensitive or problematic information from large language models without accessing their internal weights. Instead of fine-tuning the large, black-box model directly, the method learns a logit offset using a pair of smaller, white-box offset models. This approach is designed to modify the output distribution of the target model while preserving its overall performance on non-sensitive tasks. The framework is versatile, as it can be combined with various existing unlearning algorithms, and it demonstrates empirical effectiveness on a synthetic benchmark (TOFU) as well as on several general task benchmarks.

For the strength, the paper presents a novel and timely contribution to understanding LLMs (how they learn), addressing an important challenge with practical implications. The methodology is well motivated and clearly described, and the experimental results are generally sound, demonstrating consistent improvements over relevant baselines. While there are some areas that could benefit from further clarification or expanded analysis, they do not detract significantly from the overall quality and relevance of the work. The paper meets the TMLR acceptance criteria in terms of originality, soundness, clarity, and significance, and I believe it would be a valuable addition to the community. However, there are several points to be improved. While the paper presents some interesting contributions, its applicability appears to be confined to a narrow set of scenarios. The primary claim that comparable unlearning performance can be achieved without accessing the target model's weights relies on several caveats. Specifically, it requires a smaller model from the same family, access to the weights of this smaller model, and careful hyperparameter tuning for the new parameters. The authors address reviewers' concerns well in their rebuttal. Thus, I would like to recommend accept as is.

**Audience:**

Yes

**Claims And Evidence:**

Yes